# UniversalNER: TARGETED DISTILLATION FROM LARGE LANGUAGE MODELS FOR OPEN NAMED ENTITY RECOGNITION

**Wenxuan Zhou**[1*], **Sheng Zhang**[2*], **Yu Gu**[2], **Muhao Chen**[1,3], **Hoifung Poon**[2]
[1]University of Southern California   [2]Microsoft Research   [3]University of California, Davis
[1]{zhouwenx,muhaoche}@usc.edu   [2]{shezhan,yugu1,hoifung}@microsoft.com

## ABSTRACT

Large language models (LLMs) have demonstrated remarkable generalizability, such as understanding arbitrary entities and relations. Instruction tuning has proven effective for distilling LLMs into more cost-efficient models such as Alpaca and Vicuna. Yet such student models still trail the original LLMs by large margins in downstream applications. In this paper, we explore *targeted distillation* with mission-focused instruction tuning to train student models that can excel in a broad application class such as open information extraction. Using named entity recognition (NER) for case study, we show how ChatGPT can be distilled into much smaller UniversalNER models for open NER. For evaluation, we assemble the largest NER benchmark to date, comprising 43 datasets across 9 diverse domains such as biomedicine, programming, social media, law, finance. Without using any direct supervision, UniversalNER attains remarkable NER accuracy across tens of thousands of entity types, outperforming general instruction-tuned models such as Alpaca and Vicuna by over 30 absolute F1 points in average. With a tiny fraction of parameters, UniversalNER not only acquires ChatGPT's capability in recognizing arbitrary entity types, but also outperforms its NER accuracy by 7-9 absolute F1 points in average. Remarkably, UniversalNER even outperforms by a large margin state-of-the-art multi-task instruction-tuned systems such as InstructUIE, which uses supervised NER examples. We also conduct thorough ablation studies to assess the impact of various components in our distillation approach. We release the distillation recipe, data, and UniversalNER models to facilitate future research on targeted distillation.[1]

## 1 INTRODUCTION

Large language models (LLMs) such as ChatGPT (Ouyang et al., 2022; OpenAI, 2023) have demonstrated remarkable generalization capabilities, but they generally require prohibitive cost in training and inference. Moreover, in mission-critical applications such as biomedicine, white-box access to model weights and inference probabilities are often important for explainability and trust. Consequently, instruction-tuning has become a popular approach for distilling LLMs into more cost-efficient and transparent student models. Such student models, as exemplified by Alpaca (Taori et al., 2023) and Vicuna (Chiang et al., 2023), have demonstrated compelling capabilities in imitating ChatGPT. However, upon close inspection, they still trail the teacher LLM by a large margin, especially in targeted downstream applications (Gudibande et al., 2023). Bounded by limited compute, it is unsurprising that generic distillation can only produce a shallow approximation of the original LLM across all possible applications.

In this paper, we instead explore *targeted distillation* where we train student models using mission-focused instruction tuning for a broad application class such as open information extraction (Etzioni et al., 2008). We show that this can maximally replicate LLM's capabilities for the given application

---

[*] Equal contributions.
[1]Project page: `https://universal-ner.github.io/`

class, while preserving its generalizability across semantic types and domains. We choose named entity recognition (NER) for our case study, as it is one of the most fundamental tasks in natural language processing (Wu et al., 2017; Perera et al., 2020). Recent studies (Wei et al., 2023; Li et al., 2023) show that when there are abundant annotated examples for an entity type, LLMs still fall behind the state-of-the-art supervised system for that entity type. However, for the vast majority of entity types, there is little annotated data. New entity types constantly emerge, and it is expensive and time-consuming to generate annotated examples, especially in high-value domains such as biomedicine where specialized expertise is required for annotation. Trained on pre-specified entity types and domains, supervised NER models also exhibit limited generalizability for new domains and entity types.

We present a general recipe for targeted distillation from LLMs and demonstrate that for open-domain NER. We show how to use ChatGPT to generate instruction-tuning data for NER from broad-coverage unlabeled web text, and conduct instruction-tuning on LLaMA (Touvron et al., 2023a) to distill the `UniversalNER` models (`UniNER` in short).

To facilitate a thorough evaluation, we assemble the largest and most diverse NER benchmark to date (`UniversalNER` benchmark), comprising 43 datasets across 9 domains such as biomedicine, programming, social media, law, finance. On zero-shot NER, LLaMA and Alpaca perform poorly on this benchmark (close to zero F1). Vicuna performs much better by comparison, but still trails ChatGPT by over 20 absolute points in average F1. By contrast, `UniversalNER` attains state-of-the-art NER accuracy across tens of thousands of entity types in the `UniversalNER` benchmark, outperforming Vicuna by over 30 absolute points in average F1. With a tiny fraction of parameters, `UniversalNER` not only replicates ChatGPT's capability in recognizing arbitrary entity types, but also outperforms its NER accuracy by 7-9 absolute points in average F1. Remarkably, `UniversalNER` even outperforms by a large margin state-of-the-art multi-task instruction-tuned systems such as InstructUIE (Wang et al., 2023a), which uses supervised NER examples. We also conduct thorough ablation studies to assess the impact of various distillation components, such as the instruction prompts and negative sampling.

## 2 RELATED WORK

**Knowledge distillation.** While LLMs such as ChatGPT achieve promising results, these models are often black-box and have high computational costs. To address these issues, distilling the task capabilities of LLMs into smaller, more manageable models has emerged as a promising direction. Knowledge distillation (Hinton et al., 2015) often revolves around the transfer of knowledge from larger, more complex models to their smaller counterparts. Recent work (Taori et al., 2023; Chiang et al., 2023; Peng et al., 2023) seeks to distill the general abilities of LLMs with the objective of matching, if not surpassing, the performance of the original LLMs. Particularly, Alpaca (Taori et al., 2023) automates the generation of instructions (Wang et al., 2023c) and distills the knowledge from a teacher LLM. Vicuna (Chiang et al., 2023) adopts the ShareGPT data, which are comprised of real conversations with ChatGPT conducted by users, thereby providing a more authentic context for distillation. Another line of work (Smith et al., 2022; Jung et al., 2023; Hsieh et al., 2023; Gu et al., 2023) focuses on distilling task-level abilities from LLMs. Particularly, Jung et al. (2023) propose an efficient method to distill an order of magnitude smaller model that outperforms GPT-3 on specialized tasks summarization and paraphrasing in certain domains. Hsieh et al. (2022) propose to distill LLMs' reasoning abilities into smaller models by chain-of-the-thought distillation. However, these studies perform distillation either on certain datasets or domains, while our work focuses on a more general formulation that can be applied to diverse domains.

**Instruction tuning.** As an effective method to adapt LMs to perform a variety of tasks, instruction tuning has attracted an increasing number of community efforts: FLAN (Chung et al., 2022), T0 (Sanh et al., 2021), and Tk-Instruct (Wang et al., 2022) convert a large set of existing supervised learning datasets into instruction-following format, and then fine-tune encoder-decoder models, showing strong zero-shot and few-shot performance on NLP benchmarks. Ouyang et al. (2022) crowd-source high-quality instruction data and fine-tune GPT-3 into InstructGPT, enhancing its ability to understand user intention and follow instructions. Recent advancements (Taori et al., 2023; Chiang et al., 2023; Peng et al., 2023) have also led to smaller models that exhibit task-following capabilities, after being fine-tuned on instruction data generated by LLMs, such as ChatGPT or GPT4. However, these smaller

models often struggle to generate high-quality responses for a diverse range of tasks (Wang et al., 2023b). A closer examination on targeted benchmarks reveals a substantial gap between these models to ChatGPT (Gudibande et al., 2023). Our proposed method, in contrast, focuses on tuning models to excel at a specific type of tasks. The diversity in our instructing-tuning method comes from task labels (e.g., relation types for relation extraction, entity types for NER), rather than instructions. By focusing on task-level capabilities and using NER as a case study, we demonstrate that it is possible to devise a tuning recipe that not only closes the performance gap but also surpasses ChatGPT. Wang et al. (2023a) also explore instruction-tuning for information extraction tasks. However, their method relies solely on supervised datasets and yields subpar performance when compared to ChatGPT.

## 3 MISSION-FOCUSED INSTRUCTION TUNING

Instruction tuning (Ouyang et al., 2022; Wei et al., 2021) is a method through which pretrained autoregressive language models are finetuned to follow natural language instructions and generate responses. Existing work focuses on tuning models to do diverse tasks (Taori et al., 2023; Chiang et al., 2023). In contrast, we introduce a general recipe for mission-focused instruction tuning, where the pretrained model is tuned for a broad application class such as open information extraction.

In this paper, we conduct a case study on the NER task, as it is one of the fundamental tasks for knowledge extraction from text. The objective is to learn a model $f : (\mathcal{X} \times \mathcal{T}) \to \mathcal{Y}$, where $\mathcal{X}$ represents the set of inputs, $\mathcal{T}$ denotes a predefined set of entity types, and $\mathcal{Y}$ represents the set of entities of a specific type in the given input.

### 3.1 DATA CONSTRUCTION

A typical instruction-tuning example is made of three parts, including `instruction`, `input`, and `output`, where the diversity of `instruction` causes the models to follow a wide range of task instructions. However, for *mission-focused* instruction tuning, our goal is to tune the model to maximally generalize across semantic types and domains for the targeted application class. Therefore, we focus on increasing the diversity of `input` rather than `instruction`.

While earlier work (Jung et al., 2023) employs language models to generate inputs, these models typically assume that the domains of test data are known and prompt LMs to generate data for each domain. This method falls short when applied to distillation for a broad application class, where the distribution of test data is unknown. Consequently, it is challenging to generate inputs from LMs that provide wide coverage of the test domains.

To address this limitation, we propose an alternative: directly sampling inputs from a large corpus across diverse domains, and then using an LLM to generate outputs. In this paper, we sample inputs from the Pile corpus (Gao et al., 2020), which compiles 22 distinct English sub-datasets. We chunk the articles in Pile to passages of a max length of 256 tokens and randomly sample 50K passages as the inputs. Subsequently, we use ChatGPT (`gpt-3.5-turbo-0301`) to generate entity mentions and their associated types based on the sampled passages. To ensure stability, we set the generation temperature to 0. The specific prompt for constructing the data is shown in Fig. 1. In this prompt, we do not specify the set of entity types of interest, allowing the LLM to generate outputs encompassing a broad coverage of entity types.

---

**Data Construction Prompt**

**System Message:** You are a helpful information extraction system.
**Prompt:** Given a passage, your task is to extract all entities and identify their entity types. The output should be in a list of tuples of the following format: [("entity 1", "type of entity 1"), ... ].

Passage: {input_passage}

Figure 1: Data construction prompt for generating entity mentions and their types for a given passage.

---

**Data statistics.** After filtering out unparseable outputs and inappropriate entities, including non-English entities and those classified under 'ELSE' categories, such as None, NA, MISC, and ELSE, our dataset comprises 45,889 input-output pairs, encompassing 240,725 entities and 13,020 distinct entity types. We divide the entity types according to frequency and show the top 10 entity types in each range in Tab. 1. The distribution of these entity types exhibits a heavy tail, where the top

| Frequency | Entity types |
|---|---|
| Top 1% (74%) | person, organization, location, date, concept, product, event, technology, group, medical condition, ... |
| 1%-10% (19%) | characteristic, research, county, module, unit, feature, cell, package, anatomical structure, equipment, ... |
| 10%-100% (7%) | attribute value, pokemon, immune response, physiology, animals, cell feature, FAC, input device, ward, broadcast, ... |

Table 1: Examples of entities across different frequency ranges - top 1%, 1-10%, and 10-100%, along with the percentage of total frequencies for each range.

1% of entities account for 74% of total frequencies. We find that the generated data contain entity types from various domains, ranging from the general domain (e.g., PERSON) to the clinical domain (e.g., MEDICAL CONDITION). Moreover, we observe variations in granularity among the entity types. E.g., COUNTY is the subset of LOCATION, and INPUT DEVICE is a subset of PRODUCT. These data characteristics offer extensive coverage of entity types, making them suitable for distilling capabilities from LLMs across various domains.

**Definition-based data construction.** Besides entity types, we also prompt ChatGPT to generate entity mentions and define their types using short sentences. To do so, we simply change the prompt in Fig. 1 from "extract all entities and identify their entity types" to "extract all entities and concepts, and *define their type using a short sentence*". This method generates a much more diverse set of 353,092 entity types and leads to a tuned model that is less sensitive to entity type paraphrasing (Section 5.5), but performs worse on standard NER benchmarks (Section 5.2).

## 3.2 INSTRUCTION TUNING

After obtaining the data, we apply instruction tuning to smaller models to distill for a broad application class, e.g., diverse entity types in NER. Our template, as shown in Fig. 2, adopts a conversation-style tuning format. In this approach, the language model is presented with a passage $X_{\text{passage}}$ as input. Then, for each entity type $t_i$ that appears in the output, we transform it into a natural language query "*What describes $t_i$?*" Subsequently, we tune the LM to generate a structured output $y_i$ in the form of a JSON list containing all entities of $t_i$ in the passage. We consider $y_1, ..., y_T$ as gold tokens and apply a language modeling objective on these tokens. Our preliminary experiments show that conversation-style tuning is better than traditional NER-style tuning adopted by Wang et al. (2023a); Sun et al. (2023).

> ### Conversation-style Instruct Tuning Template
>
> A virtual assistant answers questions from a user based on the provided text.
> `User:` Text: $X_{\text{passage}}$
> `Assistant:` I've read this text.
> `User:` What describes $t_1$ in the text?
> `Assistant:` $y_1$
> ...
> `User:` What describes $t_T$ in the text?
> `Assistant:` $y_T$

Figure 2: The conversation-style template that converts a passage with NER annotations into a conversation, where $X_{\text{passage}}$ is the input passage, $[t_1, ..., t_T]$ are entity types to consider, and $y_i$ is a list of entity mentions that are $t_i$. The conversation is used to tune language models. Only the highlighted parts are used to compute the loss.

Besides one entity type per query, we also consider combining all entity types in a single query, requiring the model to output all entities in a single response. Detailed results and discussions can be found in Section 5.2.

**Negative sampling.** Our data construction process follows an open-world assumption where we allow the model to generate entity types that have appeared in the passage. However, the generated data do not account for entity types that are not mentioned in the passage, i.e., negative entity types. As a result, it is challenging for us to apply a model trained on this data to a closed-world setting, where one may ask for entity types that do not exist in the passage. To address this potential mismatch, we sample negative entity types from the collection of all entity types that do not appear in the passage as queries and set the expected outputs as empty JSON lists. The sampling of negative entity types

is done with a probability proportional to the frequency of entity types in the entire dataset. This approach greatly improves the instruction tuning results, as shown in Section 5.4.

**Supervised finetuning.** When we have additional human annotations, model performance can be further improved with supervised data. However, a significant challenge arises when training with multiple datasets, as there might be discrepancies in label definitions among these datasets, resulting in label conflicts. For instance, some datasets like ACE (Walker et al., 2006) consider personal pronouns (e.g., she, he) as PERSON, while other datasets like multiNERD (Tedeschi & Navigli, 2022) do not include pronouns.

To address this issue, we propose to use dataset-specific instruction tuning templates to harmonize the discrepancies in label definitions, as illustrated in Fig. 3. Specifically, we augment the input with an additional field denoting the dataset name $D$. By doing so, the model can learn the dataset-specific semantics of labels. During inference, we use the respective dataset name in the prompt for the supervised setting, whereas we omit the dataset field from the prompt in the zero-shot setting.

> **Dataset-specific Instruct Tuning Template**
>
> A virtual assistant answers questions from a user based on the provided text.
> User: Dataset: $D$ \n Text: $X_{\text{passage}}$
> Assistant: I've read this text.
> User: What describes $t_1$ in the text?
> Assistant: $y_1$
> ...
> User: What describes $t_T$ in the text?
> Assistant: $y_T$

Figure 3: The dataset-specific instruction tuning template. We add the dataset name $D$ (colored in red) as part of the input to resolve conflicts in label definitions.

## 4    UNIVERSAL NER BENCHMARK

To conduct a comprehensive evaluation of NER models across diverse domains and entity types, we collect the largest NER benchmark to date. This benchmark encompasses 43 NER datasets across 9 domains, including general, biomedical, clinical, STEM, programming, social media, law, finance, and transportation domains. An overview of data distribution is shown in Fig. 4. Detailed dataset statistics are available in Appendix Tab. 6.

**Dataset processing.** To make the entity types semantically meaningful to LLMs, we conduct a manual inspection of the labels and convert the original labels into natural language formats. For instance, we replace PER with PERSON. While we try to collect a broad coverage of NER datasets, we do not use all entity types. This is because some entity types (e.g., ELSE) are not coming from consistent sources across the different datasets. Their annotations often come from different ontologies for different purposes. The choices of entity types and their annotation guidelines are not optimized for holistic or comprehensive assessments, which renders them suboptimal for use as a "ground truth" to evaluate a universal NER model. Therefore, we remove those labels from the datasets. In addition, some datasets are at the document level and contain very long contexts, which might exceed the input length limit of models. Therefore, we split all instances in document-level datasets into sentence-level ones.

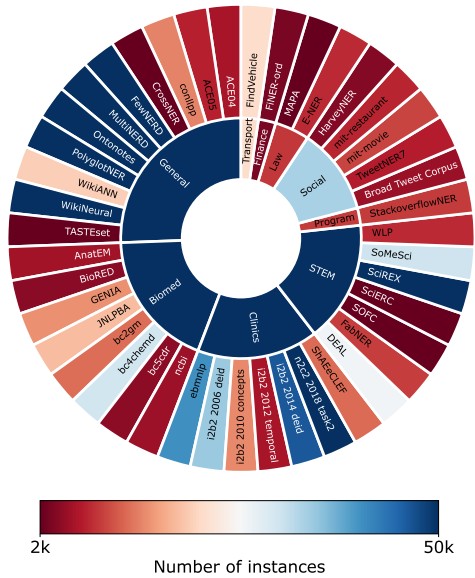

Figure 4: Distribution of UniNER benchmark.

## 5    EXPERIMENTS

This section presents experimental evaluations of `UniversalNER`. We start by outlining experimental settings (Section 5.1), followed by presenting the results on both distillation and supervised settings

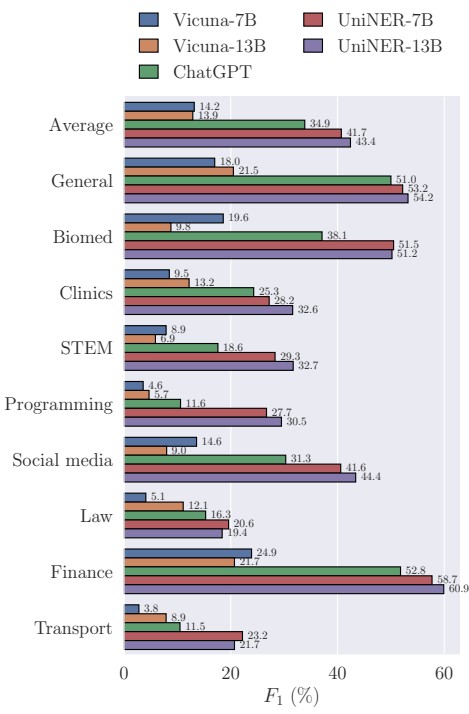

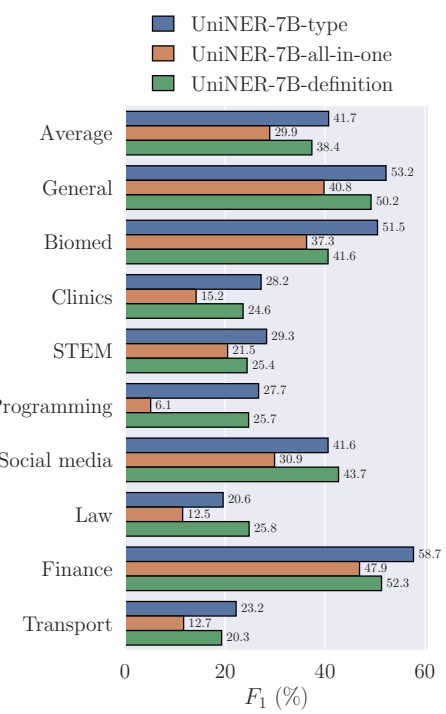

(a) Comparisons of zero-shot models on different domains. Our distilled models achieve better results than ChatGPT in all evaluated domains.

(b) Comparisons between `UniNER-7B` and two variants. `UniNER-7B-definition` is distilled on Pile data prompted with entity type definitions. `UniNER-7B-all-in-one` is tuned with the template where all entity types are asked in one query.

(Sections 5.2 and 5.3). Finally, we conduct analysis (Section 5.4) and case study (Section 5.5) to provide deeper insights into the model's performance.

## 5.1 EXPERIMENTAL SETTINGS

**Model configurations.** We train models based on LLaMA[2] (Touvron et al., 2023a) following the training schedule of Chiang et al. (2023) for a fair comparison. Considering the large size of certain test datasets, we perform evaluation by sampling up to 200,000 passage-query pairs from each dataset. We use strict entity-level micro-$F_1$ in evaluation, requiring both the entity type and boundary to exactly match the ground truth.

**Compared models.** We compare our model (`UniNER`) against the following models: (1) **ChatGPT** (`gpt-3.5-turbo-0301`). We use the prompting template in Ye et al. (2023) for NER. (2) **Vicuna** (Chiang et al., 2023) is finetuned with ChatGPT conversations, using LLaMA as the base model. (3) **InstructUIE** (Wang et al., 2023a) is a supervised model finetuned on diverse information extraction datasets, employing a unified natural language generation objective. It adopts Flan-T5 11B (Chung et al., 2022) as the base model.

## 5.2 DISTILLATION

We first evaluate the models in a zero-shot setting. We compare the performance of ChatGPT, Vicuna, and our model `UniNER`, which is distilled from ChatGPT NER annotations on Pile without human-labeled datasets in training. Results are shown in Fig. 5a.[3] We observe that our distilled

---

[2]We also train models based on LLaMA 2 (Touvron et al., 2023b). However, no significant difference is observed in our experiments.

[3]Due to limited space, we only show the average $F_1$ of all datasets and the average $F_1$ of each domain. See Appendix Fig. 9 for full results.

models, namely `UniNER-7B` and `UniNER-13B`, outperform ChatGPT in terms of average $F_1$. The average $F_1$ scores of `UniNER-7B` and `UniNER-13B` are 41.7% and 43.4%, respectively, compared to 34.9% for ChatGPT. This demonstrates that our proposed targeted distillation from diverse inputs yields models that have superior performance on a broad application class while maintaining a relatively small model size. Additionally, `UniNER-13B` exhibits better performance compared to `UniNER-7B`, indicating that fine-tuning on larger models may lead to improved generalization. In terms of domains, both `UniNER-7B` and `UniNER-13B` outperform ChatGPT on all domains, showing that the improvements exist across various domains.

We further compare different variations of `UniNER`, including (1) `UniNER-all-in-one`, where the extraction of all entity types are combined into one query and response, and (2) `UniNER-definition`, where queries in instruction tuning data use entity type definitions generated by ChatGPT instead of entity types. Results are shown in Fig. 5b. We observe that both `UniNER-all-in-one` and `UniNER-definition` underperform `UniNER-type` by 3.3% and 11.8% on average, respectively. The `UniNER-definition` variant's decreased performance could be due to its lower consistency with the evaluation datasets, which all adopt words or short phrases as labels instead of sentences. The performance disparity in the `UniNER-all-in-one` variant can be potentially attributed to the attention distribution and task complexity. When the model is required to handle multiple entity types within a single query, it might disperse its attention across these varied types, possibly resulting in less accurate identification for each individual type. Conversely, by decomposing the task into several simpler ones, each focusing on one entity type at a time, the model might be better equipped to handle the complexity, thus yielding more accurate results.

### 5.3 SUPERVISED FINETUNING

We study whether our models can be further improved using additional human annotations. We compare the performance of ChatGPT, Vicuna, InstructUIE (Wang et al., 2023a)[4], and `UniNER`.

**Out-of-domain evaluation.** We first study whether supervised finetuning leads to better generalization on unseen data. We follow InstructUIE to exclude two datasets CrossNER (Liu et al., 2021) and MIT (Liu et al., 2013) for out-of-domain evaluation, and fine-tune our model using training splits of the remaining datasets in the universal NER benchmark. Results are shown in Tab. 3. Notably, without any fine-tuning, instruction-tuned `UniNER` 7B and 13B already surpass ChatGPT, Vicuna, and the supervised fine-tuned InstructUIE-11B by a large margin. If we train our model from scratch only using the supervised data, it achieves an average $F_1$ of 57.2%. Continual fine-tuning `UniNER-7B` using the supervised data achieves the best average $F_1$ of 60.0%. These findings suggest that the models' generalization can be further improved with additional human-annotated data.

**In-domain evaluation.** We then study the performance of `UniNER` in an in-domain supervised setting, where we fine-tune `UniNER-7B` using the same training data as InstructUIE (Wang et al., 2023a). Results are shown in Tab. 2. Our `UniNER-7B` achieves an average $F_1$ of 84.78% on the 20 datasets, surpassing both BERT-base and InstructUIE-11B by 4.69% and 3.62%, respectively. This experiment demonstrates the effectiveness of our model in the supervised setting.

| Dataset | BERT-base | InstructUIE 11B | UniNER 7B |
|---|---|---|---|
| ACE05 | **87.30** | 79.94 | 86.69 |
| AnatEM | 85.82 | 88.52 | **88.65** |
| bc2gm | 80.90 | 80.69 | **82.42** |
| bc4chemd | 86.72 | 87.62 | **89.21** |
| bc5cdr | 85.28 | 89.02 | **89.34** |
| Broad Twitter | 58.61 | 80.27 | **81.25** |
| CoNLL03 | 92.40 | 91.53 | **93.30** |
| FabNER | 64.20 | 78.38 | **81.87** |
| FindVehicle | 87.13 | 87.56 | **98.30** |
| GENIA | 73.3 | 75.71 | **77.54** |
| HarveyNER | **82.26** | 74.69 | 74.21 |
| MIT Movie | 88.78 | 89.58 | **90.17** |
| MIT Restaurant | 81.02 | **82.59** | 82.35 |
| MultiNERD | 91.25 | 90.26 | **93.73** |
| ncbi | 80.20 | 86.21 | **86.96** |
| OntoNotes | **91.11** | 88.64 | 89.91 |
| PolyglotNER | **75.65** | 53.31 | 65.67 |
| TweetNER7 | 56.49 | **65.95** | 65.77 |
| WikiANN | 70.60 | 64.47 | **84.91** |
| wikiNeural | 82.78 | 88.27 | **93.28** |
| Avg | 80.09 | 81.16 | **84.78** |

Table 2: $F_1$ on 20 datasets used in Wang et al. (2023a). BERT-base results are from Wang et al. (2023a). InstructUIE results are from our reevaluation.

---

[4] Please note that the original evaluation script in InstructUIE contains a critical bug. For passages that do not contain any entities, the script adds NONE as a placeholder entity and takes it into account when calculating $F_1$. To rectify this error, we re-evaluated InstructUIE using their released checkpoint.

| Model | Movie | Restaurant | AI | Literature | Music | Politics | Science | Avg |
|---|---|---|---|---|---|---|---|---|
| *Zero-shot* | | | | | | | | |
| Vicuna-7B | 6.0 | 5.3 | 12.8 | 16.1 | 17.0 | 20.5 | 13.0 | 13.0 |
| Vicuna-13B | 0.9 | 0.4 | 22.7 | 22.7 | 26.6 | 27.2 | 22.0 | 17.5 |
| ChatGPT | 5.3 | 32.8 | 52.4 | 39.8 | **66.6** | **68.5** | **67.0** | 47.5 |
| UniNER-7B | 42.4 | 31.7 | 53.5 | 59.4 | 65.0 | 60.8 | 61.1 | 53.4 |
| UniNER-13B | **48.7** | **36.2** | **54.2** | **60.9** | 64.5 | 61.4 | 63.5 | **55.6** |
| *In-domain supervised* | | | | | | | | |
| InstructUIE-11B | - | - | 48.4 | 48.8 | 54.4 | 49.9 | 49.4 | - |
| UniNER-7B (sup. only) | 54.2 | 16.0 | 62.3 | **67.4** | 69.0 | 64.5 | 66.9 | 57.2 |
| UniNER-7B (inst-tuned + sup.) | **61.2** | **35.2** | **62.9** | 64.9 | **70.6** | 66.9 | **70.8** | **61.8** |

Table 3: Out-of-domain evaluation on datasets from Wang et al. (2023a). "sup. only" denotes a variant of UniNER-7B, trained from scratch using in-domain supervised data only and evaluated on out-of-domain datasets.

## 5.4 ANALYSIS

| Strategy | Movie | Restaurant | AI | Literature | Music | Politics | Science | Avg |
|---|---|---|---|---|---|---|---|---|
| None | 19.1 | 19.1 | 25.1 | 39.5 | 42.7 | 48.9 | 26.2 | 31.5 |
| Uniform | 42.5 | 29.0 | 42.5 | 53.3 | 57.4 | 56.8 | 52.6 | 47.7 |
| Frequency | 42.4 | 31.7 | 53.5 | 59.4 | 65.0 | 60.8 | 61.1 | 53.4 |

Table 4: Ablation study on negative sampling strategies for UniNER-7B. All models are instruction-tuned on Pile.

**Negative sampling strategies.** We experiment with different negative sampling strategies in instruction tuning, including (1) *no negative sampling*, (2) *uniform sampling* where entity types are randomly sampled with equal probability for each one, and (3) *frequency-based sampling* where we sample entity types with probabilities proportional to their frequency in the constructed dataset. Results are shown in Tab. 4. Among the approaches tested, frequency-based sampling yielded the best results, outperforming no sampling and uniform sampling by 21.9% and 5.7%, respectively. These findings highlight the crucial role of negative sampling in instruction tuning, with frequency-based sampling emerging as the most effective method for enhancing model performance in our study.

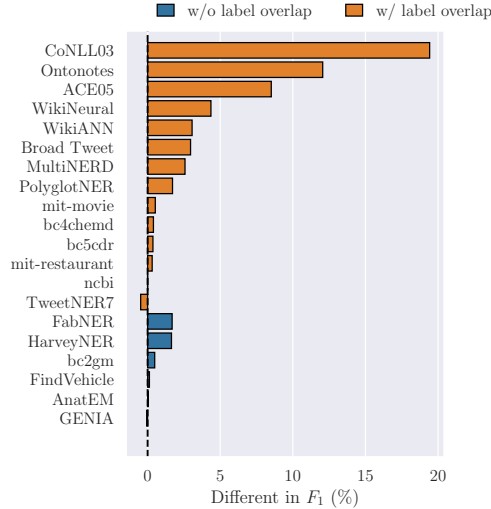

Figure 6: Different in $F_1$ between data-specific and original templates in the supervised setting. Orange and Blue mark datasets with/without label overlap with other datasets, respectively.

**Dataset-specific template.** We compare the results of our dataset-specific instruction tuning template and the original template in the supervised setting. As shown in Fig. 6, we find that the data-specific template outperforms the original template on most datasets. To gain deeper insights into the improvements achieved, we further divide the datasets into two categories: those with label (entity type) overlap with other datasets and those without overlap. Our analysis reveals that datasets with label overlap demonstrate more substantial improvements.

To explore this further, we measure $F_1$ score across all evaluation datasets and calculate the difference. Apart from the long-tail entity types that manifest a high variance in results, we identify two entity types where the dataset-specific template outperforms the original template by over 10%: FACILITY

| Partial match | Model | Movie | Restaurant | AI | Literature | Music | Politics | Science | Avg |
|---|---|---|---|---|---|---|---|---|---|
| No | ChatGPT | 5.3 | 32.8 | 52.4 | 39.8 | 66.6 | 68.5 | 67.0 | 47.5 |
| | UniNER-7B | 42.4 | 31.7 | 53.5 | 59.4 | 65.0 | 60.8 | 61.1 | 53.4 |
| | UniNER-7B w/ sup | 61.2 | 35.2 | 62.9 | 64.9 | 70.6 | 66.9 | 70.8 | 61.8 |
| Yes | ChatGPT | 5.9 | 40.1 | 55.7 | 42.8 | 70.2 | 71.7 | 70.1 | 50.9 |
| | UniNER-7B | 46.9 | 40.3 | 57.7 | 62.7 | 62.9 | 63.2 | 63.3 | 56.7 |
| | UniNER-7B w/ sup | 65.5 | 39.4 | 66.2 | 67.2 | 72.7 | 68.9 | 73.4 | 64.8 |

Table 5: Allowing partial match between the prediction and the gold that has overlap increases the results. When it is allowed, any partial match is regarded as half correct (counted as 0.5 in true positive) when computing $F_1$.

(22.0%) and TIME (12.4%). Intriguingly, both labels exhibit inconsistencies in their definitions across various datasets. The FACILITY label has been annotated on pronouns (e.g., it, which) as entities in ACE datasets but are excluded in OntoNotes. The TIME label denotes well-defined time intervals (e.g., Christmas) in MultiNERD, but may encompass any general time expressions (e.g., 3 pm) in OntoNotes. This finding suggests that the improvements provided by the data-specific template are particularly effective in resolving label conflicts.

**Evaluation with partial match.** While using strict $F_1$ as an evaluation metric, we notice that it may underestimate the zero-shot learning capabilities of NER models. In particular, strict $F_1$ penalizes slight misalignments in the boundaries of the extracted entities, which may not necessarily indicate an incorrect understanding of the text. For instance, given the sentence *any asian cuisine around* and the entity type CUISINE, UniNER extracts *asian cuisine* as the named entity, while the ground truth only labels *asian* as the correct entity. However, the model's prediction can still be viewed as correct, even though it is deemed incorrect by strict $F_1$. To better estimate the zero-shot abilities, we also consider partial match (Segura-Bedmar et al., 2013) in evaluation. In this context, a prediction that exhibits word overlap with the ground truth is regarded as half correct (counted as 0.5 in true positive) when computing $F_1$. Results are shown in Tab. 5. We find that allowing partial match consistently improves the results. Besides, our models is still the best-performing model on average.

## 5.5 Case Study

**Sensitivity to entity type paraphrasing.** One type of entity can be expressed in multiple ways, so it is essential for our model to give consistent predictions given entity types with similar meanings. An example of sensitivity analysis is present in Fig. 7. We observe that UniNER-7B-type sometimes fails to recognize entities with similar semantic meanings. On the other hand, UniNER-7B-definition, despite performing worse on our Universal NER benchmark, exhibits robustness to entity type paraphrasing. It demonstrates that although using definitions may result in lower performance on standard NER benchmarks, it could yield improved performance for less populous entity types.

**Recognition of diverse entity types.** We present an example in Fig. 8 showcasing the capabilities of UniNER in recognizing various entities. Particularly, we focus on a novel domain of code and assess UniNER's ability to extract diverse types of entities within the code. Despite minor mistakes (e.g., from_pretrained is not identified as a method), this case study effectively demonstrates our model's capacity to capture entities of various types.

## 6 Conclusion

We present a targeted distillation approach with mission-focused instruction tuning. Using NER as a case study, we train smaller and more efficient models for open-domain NER. The proposed method successfully distills ChatGPT into a smaller model UniversalNER, achieving remarkable NER accuracy across a wide range of domains and entity types without direct supervision. These models not only retain ChatGPT's capabilities but also surpass it and other state-of-the-art systems in NER performance.

ACKNOWLEDGEMENT

Wenxuan Zhou and Muhao Chen were supported by the NSF Grants IIS 2105329 and ITE 2333736.

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

## A APPENDIX

### A.1 CASE STUDY

**Sensitivity to entity type paraphrasing.** One type of entity can be expressed in multiple different ways. In this scenario, it is essential for our model to give consistent predictions given entity types with similar meanings. An example of sensitivity analysis is present in Fig. 7. We observe that UniNER-7B-type sometimes fails to recognize entities with similar semantic meanings. On the other hand, UniNER-7B-definition, despite performing worse on our Universal NER benchmark, exhibits robustness to entity type paraphrasing. It demonstrates that although using definitions may result in lower performance on standard NER benchmarks, it could yield improved performance for less populous entity types.

---

**Sensitivity Analysis of Entity Types**

Text: I'm visiting Los Angeles next week.

User: What describes city in the text?
UniNER-7B-type: ["Los Angeles"]
UniNER-7B-definition: ["Los Angeles"]

User: What describes place in the text?
UniNER-7B-type: []
UniNER-7B-definition: ["Los Angeles"]

User: What describes metropolis in the text?
UniNER-7B-type: []
UniNER-7B-definition: ["Los Angeles"]

User: What describes urban area in the text?
UniNER-7B-type: []
UniNER-7B-definition: ["Los Angeles"]

User: What describes human settlement in the text?
UniNER-7B-type: []
UniNER-7B-definition: ["Los Angeles"]

---

Figure 7: Sensitivity of UniNER-7B on different entity types. Green and red mark correct and wrong predictions, respectively.

**Recognition of diverse entity types** The code snippet provided is for text generation using the Huggingface library:

```python
from transformers import GPT2Tokenizer, GPT2LMHeadModel

tokenizer = GPT2Tokenizer.from_pretrained('gpt2')
model = GPT2LMHeadModel.from_pretrained('gpt2')

input_text = "This is a simple program"
input_ids = tokenizer.encode(input_text, return_tensors='pt')
output = model.generate(input_ids, max_new_tokens=50)
generated_text = tokenizer.decode(output[0], skip_special_tokens=True)
```

Figure 8: A code snippet to assess UniNER's ability to extract diverse types of entities within the code.

In the ensuing conversations with UniNER, we present the queries and corresponding results:

```
User: What describes library in the text?
Assistant: ["transformers"]
User: What describes local variable in the text?
Assistant: ["input_text", "input_ids", "output", "generated_text"]
User: What describes global variable in the text?
Assistant: []
User: What describes method in the text?
Assistant: ["encode", "generate", "decode"]
User: What describes class in the text?
Assistant:["GPT2Tokenizer","GPT2LMHeadModel"]
```

Despite minor mistakes (e.g., from_pretrained is not identified as a method), this case study effectively demonstrates our model's capacity to capture entities of various types.

## B  FULL EVALUATION RESULTS

Full results on ChatGPT, UniNER-7B-type, and UniNER-7B-sup+type are shown in Fig. 9.

## C  DATA STATISTICS

We show the full dataset statistics in Universal NER in Tab. 6, including the number of instances in train/dev/test data, number of entity types, average number of tokens in input text, and the average number of entities in each instance.

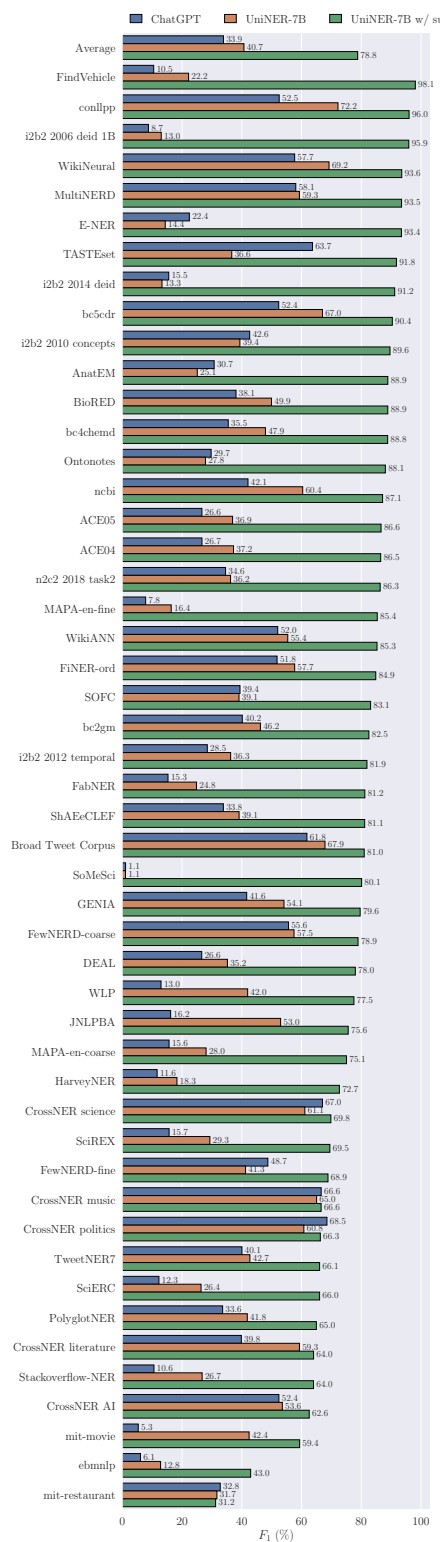

Figure 9: Full evaluation results of ChatGPT, `UniNER-7B`, and `UniNER-7B` w/ sup (joint training on both supervised and Pile-type data, MIT and CrossNER data are excluded in training).

| Domain | Dataset | # train | # dev | # test | # types | Avg. tokens | Avg. entities |
|---|---|---|---|---|---|---|---|
| General | ACE04 (Mitchell et al., 2005) | 6202 | 745 | 812 | 7 | 37 | 4.5 |
| | ACE05 (Walker et al., 2006) | 7299 | 971 | 1060 | 7 | 21 | 2.8 |
| | conllpp (Wang et al., 2019) | 14041 | 3250 | 3453 | 3 | 25 | 1.9 |
| | CrossNER AI (Liu et al., 2021) | 100 | 350 | 431 | 13 | 52 | 5.3 |
| | CrossNER literature (Liu et al., 2021) | 100 | 400 | 416 | 11 | 54 | 5.4 |
| | CrossNER music (Liu et al., 2021) | 100 | 380 | 465 | 12 | 57 | 6.5 |
| | CrossNER politics (Liu et al., 2021) | 199 | 540 | 650 | 8 | 61 | 6.5 |
| | CrossNER science (Liu et al., 2021) | 200 | 450 | 543 | 16 | 54 | 5.4 |
| | FewNERD-coarse (Ding et al., 2021) | 131767 | 18824 | 37648 | 7 | 35 | 2.6 |
| | FewNERD-fine (Ding et al., 2021) | 131767 | 18824 | 37648 | 59 | 35 | 2.6 |
| | MultiNERD (Tedeschi & Navigli, 2022) | 134144 | 10000 | 10000 | 16 | 28 | 1.6 |
| | Ontonotes (Weischedel et al., 2013) | 59924 | 8528 | 8262 | 18 | 18 | 0.9 |
| | PolyglotNER (Al-Rfou et al., 2015) | 393982 | 10000 | 10000 | 3 | 34 | 1.0 |
| | TASTEset (Wróblewska et al., 2022) | 556 | 69 | 71 | 9 | 62 | 19.1 |
| | WikiANN en (Pan et al., 2017) | 20000 | 10000 | 10000 | 3 | 15 | 1.4 |
| | WikiNeural (Tedeschi et al., 2021) | 92720 | 11590 | 11597 | 3 | 33 | 1.4 |
| Biomed | AnatEM (Pyysalo & Ananiadou, 2014) | 5861 | 2118 | 3830 | 1 | 37 | 0.7 |
| | BioRED (Luo et al., 2022) | 4373 | 1131 | 1106 | 6 | 46 | 3.0 |
| | GENIA (Kim et al., 2003) | 15023 | 1669 | 1854 | 5 | 43 | 3.5 |
| | JNLPBA (Collier & Kim, 2004) | 18608 | 1940 | 4261 | 5 | 39 | 2.8 |
| | bc2gm (Smith et al., 2008) | 12500 | 2500 | 5000 | 1 | 36 | 0.4 |
| | bc4chemd (Krallinger et al., 2015) | 30682 | 30639 | 26364 | 1 | 45 | 0.9 |
| | bc5cdr (Li et al., 2016) | 4560 | 4581 | 4797 | 2 | 41 | 2.2 |
| | ncbi (Doğan et al., 2014) | 5432 | 923 | 940 | 1 | 39 | 1.0 |
| Clinics | ebmnlp (Nye et al., 2018) | 40713 | 10608 | 2076 | 3 | 43 | 1.7 |
| | i2b2 2006 deid 1B (Uzuner et al., 2007) | 34958 | 14983 | 18095 | 8 | 16 | 0.3 |
| | i2b2 2010 concepts (Uzuner et al., 2011) | 14553 | 1762 | 27625 | 3 | 18 | 1.0 |
| | i2b2 2012 temporal (Sun et al., 2013) | 6235 | 787 | 5282 | 6 | 22 | 2.3 |
| | i2b2 2014 deid (Stubbs et al., 2015) | 46272 | 4610 | 32587 | 23 | 21 | 0.4 |
| | n2c2 2018 task2 (Henry et al., 2020) | 84351 | 9252 | 60228 | 9 | 14 | 0.6 |
| | ShAEeCLEF (Mowery et al., 2014) | 12494 | 2459 | 14143 | 1 | 13 | 0.3 |
| STEM | DEAL (Grezes et al., 2022) | 26906 | 20800 | 36665 | 30 | 35 | 1.4 |
| | FabNER (Kumar & Starly, 2022) | 9435 | 2182 | 2064 | 12 | 36 | 5.1 |
| | SOFC (Friedrich et al., 2020) | 568 | 135 | 173 | 3 | 68 | 5.3 |
| | SciERC (Luan et al., 2018) | 350 | 50 | 100 | 4 | 163 | 16.0 |
| | SciREX (Jain et al., 2020) | 71511 | 15182 | 16599 | 4 | 29 | 1.4 |
| | SoMeSci (Schindler et al., 2021) | 31055 | 159 | 16427 | 14 | 41 | 2.4 |
| | WLP (Kulkarni et al., 2018) | 8177 | 2717 | 2726 | 16 | 25 | 4.5 |
| Programming | Stackoverflow-NER (Tabassum et al., 2020) | 9263 | 2936 | 3108 | 25 | 19 | 1.2 |
| Social media | HarveyNER (Chen et al., 2022) | 3967 | 1301 | 1303 | 4 | 48 | 0.4 |
| | Broad Tweet Corpus (Derczynski et al., 2016) | 5334 | 2001 | 2000 | 3 | 28 | 0.5 |
| | TweetNER7 (Ushio et al., 2022) | 7111 | 886 | 576 | 7 | 52 | 3.1 |
| | mit-movie (Liu et al., 2013) | 9774 | 2442 | 2442 | 12 | 13 | 1.8 |
| | mit-restaurant (Liu et al., 2013) | 7659 | 1520 | 1520 | 8 | 13 | 2.2 |
| Law | E-NER (Au et al., 2022) | 8072 | 1009 | 1010 | 6 | 55 | 0.8 |
| | MAPA-coarse (Arranz et al., 2022) | 893 | 98 | 408 | 5 | 56 | 0.9 |
| | MAPA-fine (Arranz et al., 2022) | 893 | 98 | 408 | 17 | 56 | 1.3 |
| Finance | FiNER-ord (Shah et al., 2023) | 3262 | 403 | 1075 | 3 | 34 | 1.1 |
| Transportation | FindVehicle (Guan et al., 2023) | 21565 | 20777 | 20777 | 21 | 33 | 5.5 |

Table 6: Statistics of datasets in our benchmark.

