# OpenReview forum: "UniversalNER: Targeted Distillation from Large Language Models for Open Named Entity Recognition"
_ICLR.cc/2024/Conference — ICLR 2024 poster_

### Official Review · Reviewer_qcHf · 2023-10-23

**Soundness:** 2 fair
**Presentation:** 3 good
**Contribution:** 3 good
**Rating:** 6
**Confidence:** 4

**Summary:**

The authors distill a general NER model from ChatGPT data and propose a large NER benchmark. Their model seems to perform well when the types are known ahead of time.

**Strengths:**

I have read the author reply and have raised my score by one point.

----------------------

Thanks to the authors for the hard work on this paper. I have marked it a 5, but plan to raise my score if the weaknesses are adequately addressed.

Strengths:
- Good performance
- Nice benchmark contribution
- Ablations are largely good

**Weaknesses:**

- It's not clear why ChatGPT is doing worse even though you are distilling from it. You write: for "ChatGPT (gpt-3.5-turbo-0301). We use the prompting template in Ye et al. (2023) for NER." For fairness, it also makes sense to use your UniversalNER prompting template with ChatGPT. Otherwise we don't know what the cause is for you beating ChatGPT while distilling from it. Please add this experiment.
- In section 5.5, about entity type paraphrasing - this issue seems inherited from your silver training data where generated types are likely not unique while referring to the same underlying type. Seems like this issue isn't really a concern because you provide the known types from each dataset, but would be a concern for true open-domain NER where the types are not known ahead of time. No changes are requested here, but I would appreciate hearing your response to this observation.
- The section "Recognition of diverse entity types" does not "effectively demonstrate our model’s capacity to capture entities of various types." It's just a single picked example. Please consider doing a manual evaluation on 100 samples to actually demonstrate what you claim here.

**Questions:**

- In section 3.2, you write "Then, for each entity type that appears in the output, we transform it into a natural language query “What describes it?” - To be clear, here you don't ask the model to output the entity types, but you did in 3.1?
- In sec 4, Dataset processing you write "This is because some entity types (e.g., ELSE) are not coming from consistent sources across
the different datasets." Shouldn't this be addressed by the "Dataset-specific Instruct Tuning Template"?
- In figure 6, the biggest gains are for the most famous datasets. Can you try to check if these datasets have somehow been included in LLaMA's pre-training dataset?

---

> ### Author Response · Authors · 2023-11-22
> **Response to Reviewer qcHf**
>
> ### 1. Comparison with ChatGPT's Performance
> In knowledge distillation, it is not uncommon that the student model may outperform the teacher model, given that the teacher model is more general, whereas the student model just needs to focus on the specific tasks it is fine-tuned for (open NER in this case). While individual self-supervised examples may be noisy, with a large number of them, the student model may learn about the underlying cleaner patterns.
>
> For instance, consider a recent paper on knowledge distillation for drug adverse event extraction, where the student model surpasses the GPT3.5 teacher by over 6 absolute points and nearly matches the supervised performance:
>
> https://arxiv.org/abs/2307.06439
> Distilling Large Language Models for Biomedical Knowledge Extraction: A Case Study on Adverse Drug Events.
>
> Re *using the UniversalNER prompting template with ChatGPT*: We experimented with various prompting techniques for ChatGPT, selecting the most effective one to attain the strongest results. We did try the UniversalNER prompting template on ChatGPT, but it didn't perform as well, resulting in a drop of more than 4 absolute points in F1 compared to the best prompt. We ultimately used the prompt template from [Ye et al., 2023](https://arxiv.org/abs/2303.10420), which yielded the strongest ChatGPT baseline among all prompt templates we compared.
>
> ### 2. Question about Entity Type Paraphrasing
> We concur with your insight that this issue is particularly noticeable in true open-domain NER for entity types not known ahead of time. However, even for known types like "city", there may be sensitive issues, as demonstrated in Figure 8. Our hypothesis is that asking ChatGPT to generate definitions could improve recall by making it more robust to linguistic variations, while also introducing some noise that might negatively impact precision. We appreciate your suggestion and will include a discussion in our next revision.
>
> ### 3. Recognition of diverse entity types
> Thank you for your suggestion. We conducted a manual evaluation on 100 samples, which indeed confirms the UniversalNER model's ability to recognize diverse entity types across different domains.
>
> We uploaded these 100 samples to an [anonymous folder](https://drive.google.com/file/d/1ONV_2Ih4JdFgHAE-C2pqCJ30qRRw15ai/view?usp=drive_link) and also provided five samples in a separate [response](https://openreview.net/forum?id=r65xfUb76p&noteId=gYXa3fR4ex) (due to the space limit) for a quick demonstration.

---

> ### Author Response · Authors · 2023-11-22
> **Response to Reviewer qcHf (cont'd)**
>
> ### 1. Question on Entity Type Transformation:
> In Section 3.1 of our paper, our goal is to construct a diverse dataset comprising various entity types. Given that these types are identified by ChatGPT and are not known in advance, it is crucial for ChatGPT to output both the entities and their corresponding types. Conversely, in Section 3.2, the focus shifts to instruction tuning using the dataset created in Section 3.1. At this stage, as the entity types are already known and explicitly included in the query (as in "*What describes $t$* in the text??", where $t$ represents the entity type), there is no need for the model to output the entity type again. The model's task here is to respond to the query with appropriate entities related to the specified type, not to identify or output the type itself. This distinction in the model's output requirements between Sections 3.1 and 3.2 is an intentional design choice that aligns with the specific goals of each stage of our methodology.
>
> ### 2. Question on ELSE:
> Thanks for the question. The issue with ELSE and other excluded types is that they serve as catch-all types in some datasets and have no clear definition. ChatGPT won’t be able to create appropriate examples for them due to the lack of definition of these ambiguous types. Note that even if we add the dataset name, ChatGPT still won’t be able to generate appropriate examples, as it’s not privy to the idiosyncrasies of the given dataset.
>
> The dataset-specific instruction-tuning template just adds the dataset name in the prompt, which proves beneficial when supervised examples are available. This enables the model to discern fine-grained differences for the same entity type across various datasets. In the supervised setting (e.g., Table 2), we did include ELSE types. However, this won’t help the zero-shot setting, when we are using ChatGPT to generate labels.
>
> ### 3. Question on Figure 6
>  The main purpose of Figure 6 is to demonstrate the impact of employing dataset-specific instruction tuning templates on the performance, particularly in relation to label overlap. In this figure, we compare two variants of our UniNER model, both derived from LLaMA and trained on the exact same datasets. The only difference between these two models lies in using dataset-specific templates or not. Our analysis, as illustrated in Figure 6, suggests that the substantial improvement in performance for well-known datasets can be attributed to the effective reduction of label overlap through the application of dataset-specific templates. We believe that the observed improvements are not indicative of data contamination.

---

> > ### Comment · Reviewer_qcHf · 2023-11-22
> >
> > Thank you for your thorough response. I have raised my score by 1 point.

---

> ### Author Response · Authors · 2023-11-22
> **Samples of diverse entity types**
>
> We conducted a manual evaluation on 100 samples and confirmed that the UniversalNER model's ability to recognize diverse entity types across different domains. These samples were uploaded to an [anonymous folder](https://drive.google.com/file/d/1ONV_2Ih4JdFgHAE-C2pqCJ30qRRw15ai/view?usp=drive_link). Below we provided five samples for a quick demonstration:
>
> ### Example 001
> > The aircraft took off from Saddam International Airport ( later renamed Baghdad International Airport ) in Baghdad , Iraq around 11:30 p.m. , flying to Gimpo International Airport in Gangseo-gu , Seoul , South Korea , with stops at Abu Dhabi International Airport in Abu Dhabi , United Arab Emirates , and Don Mueang International Airport in Bangkok , Thailand .
>
> **User**: What describes airport in the text?
> **Assistant**: ['Saddam International Airport', 'Baghdad International Airport', 'Gimpo International Airport', 'Abu Dhabi International Airport', 'Don Mueang International Airport']
>
> ### Example 002
> > 8 8 http://cosmologist.info/cosmomc/ The BAO data sets used in this work include 1. the isotropic BAO measurements using the 6dFRS (6dF; Beutler et al. 2011 ) and SDSS main galaxy sample (MGS) (Ross et al. 2015 ) at effective redshifts and , respectively; 2. the BOSS DR12 anisotropic BAO measurements at three effective redshifts (BOSS 3 z bin) in Alam et al. ( 2017 ) or at nine effective redshifts (BOSS 9 z bin) in Wang et al. ( 2017 ) and Zhao et al. ( 2017b ); 3. the eBOSS DR14 isotropic BAO measurement at (Ata et al. 2017 ); and 4. a combination of 6dF + MGS + BOSS 3 z bin + eBOSS DR14 (All 3 z bin), or a combination of 6dF + MGS + BOSS 9 z bin + eBOSS DR14 (All 9 z bin).
>
> **User**: What describes Dataset in the text?
> **Assistant**: ['6dFRS', 'SDSS main galaxy sample', 'BOSS DR12', 'eBOSS DR14']
>
> ### Example 003
> >  For this purpose , employee benefit plans generally include profit sharing , 401 ( k ) , and 403 ( b ) plans , but do not include : IRAs ; SIMPLE , SEP , or SARSEP plans ; plans covering self-employed individuals and their employees ( formerly Keogh/H.R. 10 plans ) ; health savings accounts ; or plans investing through the Fidelity Advisor 403 ( b ) program ;  Insurance company separate accounts ;  Broker-dealer , registered investment adviser , insurance company , trust institution and bank trust department managed account programs that charge an asset-based fee ;  Current or former Trustees or officers of a Fidelity fund or current or retired officers , directors , or regular employees of FMR LLC or FIL Limited or their direct or indirect subsidiaries ( Fidelity Trustee or employee ) , spouses of Fidelity Trustees or employees , Fidelity Trustees or employees acting as a custodian for a minor child , or persons acting as trustee of a trust for the sole benefit of the minor child of a Fidelity Trustee or employee ;  Purchased for any state , county , or city , or any governmental instrumentality , department , authority or agency ;  Purchased by a charitable organization ( as defined for purposes of Section 501 ( c ) ( 3 ) of the Internal Revenue Code ) or a charitable remainder trust or life income pool established for the benefit of a charitable organization ;  Qualified tuition programs for which FMR or an affiliate serves as investment manager , or mutual funds managed by Fidelity or other parties ;  public and private retirement programs and non-U.S. insurance companies , if approved by Fidelity ;  Destiny Planholders who exchange , or have exchanged , from Class O to Institutional Class of Fidelity Advisor funds .
>
> **User**: What describes business in the text?
> **Assistant**: ['FMR LLC', 'FIL Limited', 'Fidelity']
>
> ### Example 004
> > The top selling hybrids in the U.S . market are the Fusion Hybrid with 127,572 units , followed by Escape Hybrid with 117,997 units , and the C-Max Hybrid with 54,236. , Ford is the world 's second largest manufacturer of hybrids after Toyota Motor Corporation , with 400,000 hybrid electric vehicles produced since their introduction in 2004 .
>
> **User**: What describes car in the text?
> **Assistant**: ['Fusion Hybrid', 'Escape Hybrid', 'C-Max Hybrid']
>
> ### Example 005
> > The city 's parks department also operates two Par 4 18-hole public golf courses , 1 Par 3 18-hole course , one Par 4 9-hole golf course , two disc golf courses , Garvin Park , Lloyd Pool , the Goebel Soccer Complex , Swonder Ice Arena and the C.K .
>
> **User**: What describes sports facility in the text?
> **Assistant**: ['Lloyd Pool', 'Goebel Soccer Complex', 'Swonder Ice Arena']
>
> We will include these examples in our final version.

---

### Official Review · Reviewer_YLpo · 2023-10-28

**Soundness:** 3 good
**Presentation:** 3 good
**Contribution:** 2 fair
**Rating:** 3
**Confidence:** 5

**Summary:**

The paper explores targeted distillation with mission-focused instruction tuning to train student models that excel in open information extraction. The case study focuses on named entity recognition (NER), demonstrating how ChatGPT can be distilled into smaller UniversalNER models for open NER. UniversalNER achieves remarkable NER accuracy across tens of thousands of entity types, outperforming general instruction-tuned models like Alpaca and Vicuna by over 30 absolute F1 points on average. Furthermore, UniversalNER outperforms state-of-the-art multi-task instruction-tuned systems like InstructUIE, even without using supervised NER examples.

**Strengths:**

- The paper proposed a novel approach to distill LLMs into more cost-efficient models.
- The experiments are sufficient. The writing and presentation are clear and easy to read.

**Weaknesses:**

- Lack of some supervised baselines such as [a]. The “BERT” column in Table 2 did not represent the SoTA supervised baselines. Even so, the presented results in Table 2 show that the UniNER with such a large scale of parameters did not improve significantly and are not comparable with supervised small models in some dataset. Therefore, it is doubtful to use prompt-based method with LLM to solve NER problems. Moreover, the training cost are not presented explicitly. The motivation should be clarified carefully.
- The proposed method could be very unstable as the dataset construction includes several uncontrol factors. Thus the reproduction of the results could be problematic.  I suggest the authors should claim the related limitations clearly.

[a] Liu et al. Autoregressive Structured Prediction with Language Models. 2022

**Questions:**

- How to control the data quality constructed by ChatGPT?
- The collecting of the passage-query pairs of instruction tuning is not detailed enough for reproduction.

---

> ### Author Response · Authors · 2023-11-22
> **Response to Reviewer YLpo**
>
> ### 1. Re Supervised Baselines
> Thank you for suggesting an additional supervised baseline. We will include it and add discussion in the final version. However, we do want to point out that the suggested baseline from ([Liu et al. 2022](https://arxiv.org/abs/2210.14698)) was *only evaluated on CoNLL and ACE-05, which are classic closed NER datasets*. In contrast, our evaluation in Table 2 is based on a much broader collection of datasets introduced by the latest InstructUIE paper ([Wang et al. 2023a](https://arxiv.org/abs/2304.08085)), which we compared against and demonstrated a substantial performance gain.
>
> In sum, the supervised results in the suggested [Liu et al. 2022](https://arxiv.org/abs/2210.14698) paper were attained on much easier datasets with substantially narrower scope. They are not comparable to Table 2, which reported results on much more challenging datasets. UniNER attains the state-of-the-art accuracy and represents a large step forward compared to prior state of the art InstructUIE.
>
> In general, it’s also important to emphasize that the true power for UniNER lies in **open** NER in the **zero-shot setting**, where no manual annotation or supervised labels are available. E.g., see Figure 5. This open NER setting is an important research frontier, as many real-world scenarios require recognition capabilities for a large number of entity types, for which supervised learning simply can not scale. E.g., in biomedicine, there are tens of thousands of diagnostic types alone, not to mention other medical concepts.
>
> There may exist supervised baselines for a tiny fraction of the entity types, where there are prior efforts in annotating labeled data for supervised learning. However, for the vast majority of entity types in real-world applications, it’s simply infeasible to train supervised NER systems covering all of them, which would incur prohibitive cost in annotating sufficient number of labeled examples for each entity type. We agree that UniversalNER is far from perfect for many entity types, but as a zero-shot open NER system, it attains the state-of-the-art performance, and represents a big leap forward compared to existing generalist NER systems.
>
> Similarly, it’s important to note that any training cost in data generation and distillation is dwarfed by the efforts to train supervised NER systems for all possible entity types.
>
> ### 2. Re Instability
> To address potential instability in our data construction process, we set the `temperature` parameter to 0 during the data generation phase with GPT-3.5-turbo. Our initial exploration confirmed that this adjustment (i.e., `temp=0`) ensured consistent and reproducible outputs for our dataset. This approach mitigates the variability often associated with generative models and enhances the stability of our dataset, providing a solid foundation for our experiments.
>
> We understand the importance of reproducibility in scientific research and are committed to facilitating it for our work. To this end, we plan to release the pretraining data and the necessary scripts for reproducing pretraining and inference, enabling other researchers to independently verify the effectiveness of UniversalNER. Additionally, our repeated re-training of the model has consistently reproduced the reported results. We believe these measures effectively address the reviewer's concerns regarding stability and reproducibility, and we hope this clarification demonstrates our commitment to maintaining the highest standards of scientific rigor in our work.
>
> ### 3. Re Collection of Passage-Query Pairs
> Regarding the question about the collection of passage-query pairs, we followed a standard practice to ensure a balanced representation of datasets. Specifically, for each dataset containing more than 200,000 passage-query pairs, we subsampled 200,000 pairs to include in the instruction-tuning set. This approach prevents larger datasets from dominating the learning process. We will release our subsampled datasets to facilitate reproduction and ensure transparency.

---

### Official Review · Reviewer_rYsM · 2023-11-01

**Soundness:** 3 good
**Presentation:** 3 good
**Contribution:** 3 good
**Rating:** 8
**Confidence:** 4

**Summary:**

This paper proposed a targeted distillation model for open-named entity recognition. It also assembled a large and diverse NER benchmark with 43 NER datasets.

**Strengths:**

Although the knowledge distillation from powerful LLM such as ChatGPT is not a novel idea, this paper explored distillation and further improvement. The dataset constructed in this paper is essential and I would love it will be released soon.  Experiment results demonstrated the effectiveness of the proposed model, and the UniNER is better than ChatGPT on open domain NER, making the UniNER a better choice for local deployment for NER tasks.

**Weaknesses:**

Some details were missing in this paper, which may reduce the reproducibility of this research. For example, the dataset processing step has a filter on entity type; how to select the entity type ( the exact standard) and the entity types removed from each dataset need to be released as an appendix.

**Questions:**

None

---

> ### Author Response · Authors · 2023-11-22
> **Response to Reviewer rYsM**
>
> Thank you for the valuable and insightful feedback. We appreciate your recognition of the importance and potential impact of the dataset we constructed. Please find our response regarding your concerns on reproducibility and dataset processing below.
>
> ### 1. Entity Type Selection and Removal
> During the data preprocessing phase, we took several steps to ensure the quality and consistency of our dataset:
> - We excluded any unparsable outputs identified by JSON parsing errors.
> - We removed non-English entities and their associated types to maintain a consistent linguistic framework.
> - We omitted entities classified under 'NA' types, such as `none`, `NA`, and `else`, since they are either ambiguous or undefined, hindering effective analysis.
>
> ### 2. Handling of 'ELSE' and Syntactic Types
> We systematically remove all 'ELSE' types (e.g., `else`, `NA`, `O`, `none`, `misc`) from our dataset. This decision is based on the unsuitability of these types for zero-shot evaluation, due to their broad and often non-specific nature. Additionally, the WLP dataset contains semantic role labels (e.g., modifier, action), which we exclude to maintain a focus on semantically rich and contextually relevant entities.
>
> ### 3. Reproducibility and Data Release
> We appreciate that the reviewer recognized our dataset as a major contribution. We plan to release it, along with the pretrained models and scripts necessary for reproducing pretraining and inference, upon paper acceptance. provide details on dataset processing in the appendix and include a list of the removed entity types.

---

### Meta-Review · Area_Chair_hf5k · 2023-12-13

**Metareview:**

This paper presents a targeted distillation technique for training an open-domain NER system from data generated by an LLM. The paper also presents newly collated large benchmark for open NER. The distilled model achieves state-of-the-art results on the benchmark, outperforming even the appropriately-prompted teacher LLM as well as other state-of-the-art open NER baselines. Reviews were generally positive, finding value in the effectiveness and simplicity of the proposed method, as well as the new benchmark and generated training data. Some minor weaknesses were discussed -- for example, some missing details in pre-processing. One reviewer was more negative, but their major criticism -- that some highly-performant supervised baselines were missing -- was handled well in rebuttal: the authors pointed out that the system in question is a closed NER system and was only evaluated on CoNLL.

**Justification For Why Not Higher Score:**

While the resulting system has great practical utility and the value of the newly collated benchmark is clear, the results of this study (that an LLM teacher can be effectively distilled into a smaller targeted NER model, surpassing the teacher at NER) are not entirely unexpected.

**Justification For Why Not Lower Score:**

The biggest weakness brought up -- that a performant supervised baseline was missing from comparisons -- was adequately addressed by authors in rebuttal: the baseline in question is for closed NER, and was only evaluated on CoNLL, a much easier NER benchmark than considered in this paper.

---

### Decision · Program_Chairs · 2024-01-16

Accept (poster)